# A Significant Increasing Risk Association between Cigarette Smoking and *XPA* and *XPC* Genes Polymorphisms

**DOI:** 10.3390/genes14071349

**Published:** 2023-06-27

**Authors:** Safiah Almushawwah, Mikhlid H. Almutairi, Abdullah M. Alamri, Abdelhabib Semlali

**Affiliations:** 1Biochemistry Department, College of Science, King Saud University, P.O. Box 2455, Riyadh 11451, Saudi Arabia; 2Zoology Department, College of Science, King Saud University, P.O. Box 2455, Riyadh 11451, Saudi Arabia; 3Groupe de Recherche en Écologie Buccale, Faculté de Médecine Dentaire, Université Laval, 2420 Rue de la Terrasse, Local 1758, Québec, QC G1V 0A6, Canada

**Keywords:** smoking, nucleotide excision repair, *XPA*, *XPC*, single nucleotide polymorphism, Taq man genotyping

## Abstract

Cigarette smoking (CS) is a major cause of various serious diseases due to tobacco chemicals. There is evidence suggesting that CS has been linked with the DNA damage repair system, as it can affect genomic stability, inducing genetic changes in the genes involved in the repair system, specifically the nucleotide excision repair (NER) pathway, affecting the function and/or regulation of these genes. Single nucleotide polymorphism (SNP), along with CS, can affect the work of the NER pathway and, therefore, could lead to different diseases. This study explored the association of four SNPs in both *XPA* and *XPC* genes with CS in the Saudi population. The Taq Man genotyping assay was used for 220 healthy non-smokers (control) and 201 healthy smokers to evaluate four SNPs in the *XPA* gene named rs10817938, rs1800975, rs3176751, and rs3176752 and four SNPs in the *XPC* gene called rs1870134, rs2228000, rs2228001, and rs2607775. In the *XPA* gene, SNP rs3176751 showed a high-risk association with CS-induced diseases with all clinical parameters, including CS duration, CS intensity, gender, and age of smokers. On the other hand, SNP rs1800975 showed a statistically significant low-risk association with all clinical parameters. In addition, rs10817938 showed a high-risk association only with long-term smokers and a low-risk association only with younger smokers. A low-risk association was found in SNP rs3176752 with older smokers. In the *XPC* gene, SNP rs2228001 showed a low-risk association only with female smokers. SNP rs2607775 revealed a statistically significant low-risk association with CS-induced diseases, concerning all parameters, except for male smokers. However, SNP rs2228000 and rs1870134 showed no association with CS. Overall, the study results demonstrated possible significant associations (effector/and protector) between CS and SNPs polymorphisms in DNA repair genes, such as *XPA* and *XPC,* except for rs2228000 and rs1870134 polymorphisms.

## 1. Introduction

In the 1930s, epidemiologists used case-control surveys to investigate the relationship between lung cancer and smoking. The first study was published in 1939 by Müller, and this study shows that the incidence of lung cancer was higher in smokers than in non-smokers [1,2]. Over the years, the number of smokers in the world has increased, and many various serious diseases have been attributed to smoking, such as cardiovascular diseases [3], obstruction of the respiratory tract, age-related macular degeneration, and the risk of developing diabetes, which is all 30–40% higher for smokers than nonsmokers. Additionally, many studies showed that there are different types of cancer that are related to CS, such as lung, oral cavity, larynx, tongue, bladder, esophagus, colon, and rectum cancers [4]. According to the World Health Organization, smoking was found to be linked with more than 6,000,000 million deaths cases every year and is estimated to cause 80% of lung cancer deaths [5]. The incidence of CS in certain regions in the Kingdom of Saudi Arabia is greater than 50% [6]. Therefore, a better understanding of the potential association between CS and relative diseases is critical. Recently, many researchers have demonstrated that CS induces DNA damage and has an impact on the genes that are critical in controlling the cell cycle, such as P53 and KRAS. Mutation in both genes was found to be linked with smoking and various types of cancers [7]. Other genetic analyses of many cancers caused by smoking reveal a number of mutations, particularly in cancer-associated genes, such as GLUT-1, HIF-1 [8]. Furthermore, evidence suggests smoking to be linked with the DNA damage repair system, as it can induce genetic changes involved in the repair system, affecting the function and/or regulation of these genes [9,10,11]. Damage is detected by a DNA repair system, which is critical for maintaining the integrity and stability of genomic DNA of the cell. Any deficiency in the DNA repair genes causes DNA damage, as well as mutation, leading to diseases [12]. The nucleotide excision repair (NER) is the primary pathway responsible for removing a lesion from bulky DNA adducts caused by chemicals in tobacco smoke. In mammalian cells, over 20 protein factors constitute this critical pathway, including XPA replication protein A (RPA), which is essential for maintaining DNA stability during the repair process, as well as XPC-hHR23B- centrin 2 (CETN2), which is crucial in recognizing DNA damage in the first step of the repair process. In the current study, we will focus on the genes that belong to the XPA and XPC proteins. Furthermore, other proteins are fundamental in the NER pathway, including TFIIH, XPB, and XPD DNA helicases, as well as ERCC1-XPF and XPG [10,13,14]. XPB and XPD helicases are key members of the human TFIIH complex, and they have a critical role in unwinding DNA for the repair of duplex distorting damage by NER [15]. Modifications in NER genes will lead to many diseases. Studies established that a defect in the NER pathway gene causes Xeroderma Pigmentosum (XP) [16]. It was clearly documented that the ERCC-1 gene is closely linked to sensitivity to cisplatin therapy [17].

The xeroderma pigmentosum group C (XPC) is a 940-residue multi-domain protein. The C-terminal portion of XPC (492–940; XPC-C) is important in the interactions with DNA, TFIIH, RAD23B, and CETN2. Biochemical and structural analyses confirmed that the XPC complex is capable of binding specifically to DNA damage sites associated with a relatively bulky distortion of the DNA duplex [18,19]. XPA is an acronym for XP complementation group A (XPA). The recessive genetic disorder of XP in children leads to extreme sensitivity to UV light, which causes skin cancer (Bowden et al., 2015). XPA is involved in the NER pathway and has a critical role in the verification of DNA damage and mobilization of repair proteins. Unlike other proteins in the NER pathway, XPA is unique, as it is required for both GG-NER and TC-NER [20,21]. This research proposal was focused on the genetic variation effects of CS on DNA repair system dysfunction, resulting in health problems.

The specific aim was to investigate the polymorphism variations in genes of the NER pathway caused by CS in Saudi smokers versus non-smoker subjects.

## 2. Materials and Methods

### 2.1. Ethics Statement and Samples Collection

Written ethical consent was already reviewed and obtained by the Research Ethics Committee of the College of Applied Medical Sciences at King Saud University (KSU) in Riyadh, Saudi Arabia (Approval Number: CAMS 13/3536). As described in our previous work [6], smokers were divided into two groups based on daily quantity of cigarette consumption (≥10 and <10 cigarettes/day). All volunteer participant in the current study signed written informed consent. Clinical data on smoking history, allergic symptoms and diseases, number of cigarettes smoked daily, and body mass index (BMI) were obtained through a self-completed questionnaire. The control group corresponds to non-smokers, and the former smokers were excluded from the control group. Additionally, as described in our previously study [22], all samples were collected from self-reported healthy smokers and non-smokers (controls) who had signed informed consent forms, confirming their participation in the present study. We excluded any potential participants who self-reported having symptoms, such as metabolic disorders, inflammatory diseases, autoimmune diseases, cancer, or blood diseases. The blood samples (3 mL) were collected via venipuncture in EDTA-containing tubes. A total of 421 participants were divided into 220 healthy non-smokers (control) and 201 healthy smokers. Baseline characteristics of cases and controls were performed (Age, no. of CS/day ≤ 10 sticks per day (moderate) and >10 sticks per day as heavy smokers, and period of smoking). In addition, it was made sure that they had not taken any other drug and that they had no disease. Samples of participants who did not meet the criteria were excluded from the study.

### 2.2. DNA Extraction

Genomic DNA was extracted from 200 μL of EDTA anticoagulated peripheral blood using Qiagen QIAamp^®^ DNA Mini Kit (Q), as per the manufacturer’s instructions. The DNA concentration quantity was determined using a NanoDrop 8000 (Thermo Fisher Scientific, Waltham, MA, USA). The purity of each DNA sample was then determined by calculating the ratio of A260/A280 nm and A260/A230 nm. The samples were considered contamination-free when the ratios were 1.7–2.0. The DNA samples were preserved at −20 °C.

### 2.3. SNP Selection Taq Man Genotyping Assay

A Taq Man assay was carried out for all 421 samples to examine the polymorphism variation of *XPA* and *XPC* genes. The DNA blood samples were diluted to obtain a final concentration of 10 ng/µL. Four SNPs in the *XPA* gene, named rs10817938, rs1800975, rs3176751, and rs3176752, as well as four SNPs in the *XPC* gene, called rs2607775, rs2228000, rs2228001, and rs1870134, were evaluated by the genotyping assay. The selection of these SNPs was based on the previous review, and they were selected from the NCBI database (http://www.ncbi.nlm.nih.gov/snp, accessed on 1 February 2023). Each SNP was located either in promoter, intron, or exon regions (Appendix A), which may lead to a change in the regulation of the protein or modification in protein folding or function. As described by our previous works [23,24,25,26], a total of 8 µL of the final reaction mix and 2 µL of DNA (10 ng/µL) were distributed in an optical reaction plate. The SNP reaction mix contained 5.3 µL of TaqMan^®^ genotyping master mix (Applied Biosystems, Foster City, CA, USA), 2.5 µL of nuclease-free water, and 0.2 µL of SNP. The negative control (no DNA) was included in each plate. PCR amplification was carried out under the following conditions: a primary denaturation step at 95 °C for 7 min, followed by 40 cycles of 95 °C for 30 s, 60 °C for 1 min, and 72 °C for 30 s. The PCR reaction mixture was terminated for a final extension at 72 °C for 5 min. Quant Studio™ 7 Flex Real-Time PCR System (Applied Biosystems) with an endpoint reading of the genotypes was used to perform the reaction.

### 2.4. Statistical Analysis

Statistical analysis was carried out using the Statistical Package for Social Sciences (SPSS) version 26.0 (IBM-SPSS, Armonk, NY, USA). Hardy Weinberg equilibrium was used to check the deviation of the computed genotypic and allelic frequencies of each SNP. Genetic comparisons were performed with the aid of the χ2 test and allelic odds ratios (ORs). The chi-square test was used to determine the proportion of cases versus control and according to different groupings (based on duration of smoking, frequency of smoking, gender, and age for the SNPs and alleles). Odds ratio and 95% confidence interval were calculated using the online OR calculator (Medcalc, https://www.medcalc.org/calc/odds_ratio.php, accessed on 1 February 2023). In addition, Fisher’s exact test (two-tailed) was applied. Results were expressed as the mean and the standard deviation for age. The proportion of SNPs and alleles were reported as numbers and percentages. An independent sample t-test was performed to compare the mean age between smokes and non-smokers. *p* values of less than 0.05 were considered statistically significant. OR more than one indicates high-risk association, and less than one indicates a low-risk association.

## 3. Results

### 3.1. Baseline and Clinical Characteristics of Participants

A total of 220 Saudi healthy non-smokers and 201 healthy smokers were used in the study. Participants were further classified into different groups based on age, gender, period of smoking, and the number of CS per day. Table 1 illustrates the baseline and clinical characteristics of participants measured for all participants. These variable parameters were used to study the association between SNPs in tested genes and the risk of CS causing disease.

### 3.2. Global Genotyping Analysis of XPA and XPC among Smokers and Non-Smokers

To evaluate the association of SNPs in the *XPA* and *XPC* genes with the effects of CS on induced diseases, we evaluated rs10817938, rs1800975, rs3176751, rs3176752, rs1870134, rs2228000, rs2228001, and rs2607775 variants and CS in 421 participants. The distributions of genotyping and allele frequencies of the smoker and non-smoker groups are summarized in Table 2. 

For SNP rs10817938 (T/C) of *XPA* gene, no association was found with the risk related to smoking induced diseases. In SNP rs1800975 of the *XPA* gene, the genotyping distribution was as follows: 96.4% TT, 3.1% TC, and 0.5% CC in smokers, while it was 55.1% TT, 6.5% TC, and 38.4% CC in non-smokers. The TC, CC, and TC+CC alleles of rs1800975 decreased the risk of developing diseases related to smoking by approximately 72.6%, 99.2%, and 91.9% compared to the TT homozygous allele [TC (OR = 0.274; CI = [0.103–0.733]; *p* = 0.009); CC (OR = 0.008; CI = [0.001–0.056]; *p* < 0.001; and TC+CC (OR = 0.081; CI = [0.037–0.178]; *p* < 0.001]. The phenotypic distribution was 97.9% T and 2.1% C in smokers and 58.3% T and 41.7% C in controls with a protective affect (OR = 0.029; CI = [0.014–0.061]; *p* < 0.001).

SNP rs3176751 exhibited significant differences between smoker and non-smokers with a higher risk of smoking-induced diseases. The genotyping distribution was as the follows: 58% GG and 66.7% CG+GG in smokers, while it was 19.6% GG and 31.1% CG+GG in non-smokers when compared to the CC reference genotype (GG: OR = 6.105, CI = [3.869–9.631], *p* < 0.001; CG+GG = OR = 2.147, CI = [1.511–3.050], *p* < 0.001). The C allele was used as a reference. The G allele was found to be more frequent in smokers (62.3%) and in the non-smoker group (25.3%) compared to the C allele. The G allele showed significant high-risk association with smoking-induced diseases among smokers, as shown in Table 2 (OR = 4.869, CI = [3.618–6.555], *p* < 0.001). Additionally, SNP rs3176752 of the XPA gene did not show any association with CS. The genotyping distribution of GG, GT, and TT variants was estimated to be 97%, 3%, and 0%, respectively, in smokers, and it was 96%, 4%, and 0%, respectively, in the controls.

We have evaluated the association SNPs in the *XPC* gene (rs1870134, rs2228000, rs2228001, and rs2607775) variations and CS in 421 participants. The data for three SNPs, rs1870134, rs2228000 and rs2228001, did not show any association with the increase or decrease in the risk of smoking-induced diseases in the Saudi population. However, SNP rs2607775 exhibited significant differences between smokers and non-smokers, presenting a high risk of smoking-induced diseases in the CC genotype and the C allele. The genotyping distribution was as follows: 20.1% CC in smokers and 8.8% CC in non-smokers; CC: OR = 2.810, CI = [1.509–5.233], *p* = 0.001; the G allele was used as a reference. The C allele was found to be more frequent in smokers (39.2%) and in the non-smoker group (28.7%) compared to the G allele. The C allele showed significant high-risk association with smoking-induced diseases among smokers, as shown in Table 3 (OR = 1.598, CI = [1.192–2.143], *p* = 0.002).

### 3.3. The Association between SNPs in XPA, XPC, CS Duration, Daily CS Average, Gender, and Age

The smokers and non-smokers were divided into different groups based on CS duration, daily CS average gender, and age. The associations of the four SNPs in XPA and the four SNPs in *XPC* genes with clinical characteristics were evaluated. Table 3 and Table 4 compare genotyping and allele frequencies for each SNP in *XPA* and *XPC* genes based on different clinical characteristics.

#### 3.3.1. Estimation for SNPs XPA (rs10817938, rs1800975, rs3176751, and rs3176752)

To evaluate the relationships between different *XPA* rs10817938 genotypes and CS in the control and smokers, we distributed the study based on smoking duration (short-term smokers ≤ five years and long-term smokers > five years), frequency of smoking (≤10 times and >10 times), gender (males and females), and the average age of smokers (≤28 years and >28 years). The results of genotype and allele distributions of the rs10817938 variant in smokers and controls, with its different clinical characteristics, are described in Table 3A–D. *XPA* rs10817938 showed an association only with the duration of smoking (smoking for ≤five years and >five years) in the TC+CC genotype. The distribution of genotyping frequency was 37.8% and 35.7%. In short-term and long-term smokers, only the TC+CC genotype showed low-risk association (TC+CC: OR = 0.402, CI = [0.254–0.637], *p* < 0.001 for short -term; while, in long-term-smoking: OR = 0.380, CI = [0.239–0.603], *p* < 0.001).

SNP rs1800975 has a significant association with all clinical parameters. With regards to smoking duration (more/less five years), the genotyping and allele distribution were analyzed in Table 3A. In a period of smoking of more than five years, the results showed highly significant low-risk association in genotype TC compared to the TT genotype. The heterozygous TC genotype in short-term smoking exhibits 0.202-fold low-risk effects in smokers (OR = 0.202; CI = [0.045–0.914]; *p* = 0.038). This went along with the CC genotype, which posed a 0.08-fold low-risk effect in short-term smokers (OR= 0.084; CI = [0.072–0.994]; *p* < 0.049) and long term-smoking (OR = 0.019; CI = [0.001–0.145]. Allele C was shown to be significant with regards to protection from the effects of smoking (OR = 0.034; CI = [0.012–0.093]; *p* < 0.001), (OR = 0.033; CI = [0.012–0.090]; in long term smoking and short-term smoking, respectively). The TC+CC genotype showed a reduced risk of association by 92% and 89.4% in both short- and long-term smoking (OR = 0.077; CI = [0.024–0.249]; *p* < 0.001) (OR = 0.106; CI = [0.037–0.297]; *p* < 0.001). The CC genotype presented a significant correlation with protection effects of moderate smokers (OR = 0.016; CI = [0.002–0.14]; *p* < 0.001). The *C. allele* posed 0.050 low-risk effects with moderate smokers (OR = 0.050, CI = [0.020–0.124] and *p* < 0.001). Additionally, TC+CC showed low risk association with moderate smoking (OR = 0.125; CI = [0.059–0.389]; *p* = 0.001). Heavy smokers showed significant low-risk association with the TC, CC, and TC+CC genotypes (OR = 0.0185 TC, 0.008 CC and 0.047 TC+CC); (CI = [0.041–0.834], *p* = 0.028); (CI = [0.001–0.126], *p* < 0.001); (CI = [0.011–0.0196], *p* < 0.001). In Table 3C, males showed significant low-risk association with smokers. The CC variant, which is homozygous in males, exhibited significant low-risk association with smokers (OR = 0.008; CI = [0.001–0.059]; *p* < 0.001), along with the *C. allele* (OR = 0.019; CI = [0.008–0.018]; *p* < 0.001). Additionally, the *C. Allele* presented 0.168-fold protective effects in female smokers (OR = 0.0.168; CI = CI = [0.050–0.565]; *p* < 0.004). The SNPs showed low-risk association with age for both those under and above 28 years, as shown in Table 3D. The heterozygous TC presented a significant protective correlation with older smokers (OR = 0.3372; CI = [0.11809–1.965012]; *p* = 0.043). The CC homozygous variant genotype has a low-risk relationship with smokers in subjects older than 28 years (OR = 0.216; CI = [0.124–0.378); *p* < 0.001). Additionally, the *C. allele* showed a significant low-risk association with smokers older than 28 (OR = 0.234; CI = [0.161–0.341; *p* < 0.001). In contrast, the TC variant of smokers under 28 years showed no significant association. The CC variant (homozygous) of smokers under 28 years also showed a low-risk relationship with smokers (OR = 0.533010; CI = [0.357–0.079]; *p* < 0.002). Furthermore, there a low-risk association was observed in the *C. allele* (OR = 0.0556; CI = [0.424–0.7290]; *p* < 0.001).

The rs3176751 SNPs showed significant high-risk association with all clinical parameters tested in this study. The genotyping and allele distribution were shown in Table 3A–D. For example, the allele frequencies analysis of the GG genotype and the *G. allele* showed that short-term smokers (≤ five years) and long-term smokers (>five years) revealed high-risk significant association when compared to non-smokers (for short-term smokers: (OR = 7.866 GG and 6.150 G), (CI = [(4.4012–14.056)], *p* < 0.001), (CI = [4.197–9.013], *p* < 0.001), respectively; for short-term smokers: (OR = 5.691 GG and 4.529 G), (CI = [3.207–10.099], *p* < 0.001); (CI = [3.111–6.593], *p* < 0.001), respectively). The moderate and heavy smokers both displayed a significant high-risk relationship when compared to non-smokers (Table 3B). The GG genotype presented a significant high-risk association of moderate smokers (OR = 7.692; CI = [4.148–14.262]; *p* < 0.001). The *G. allele* presented 6.015 high-risk effects with moderate smokers (OR = 6.015, CI = [4.004–9.035] and *p* < 0.001). Heavy smokers showed a significant high-risk association with the GG genotype and the *G. allele* ((OR = 6.230 GG, 4.909 G); (CI = [3.575–10.858] *p* < 0.001); (CI = [3.415–7.058], *p* < 0.001), respectively). The combination of CG+GG has a significant high-risk association in both moderate and heavy smokers (*p* < 0.001 (OR = 2.294, 2.180; CI = [1.466–3.590], CI = [1.438–3.305], respectively). In addition, there was significant high-risk association for both genders, male and female, as well as the age of subjects for the GG genotype and the *G. allele,* as shown in Table 3C, D. The GG genotype presented a significant high-risk association of male smokers (OR = 5.349; CI = [3.372–8.482]; *p* < 0.001). The *G. allele* presented 4.308 high-risk effects with male smokers (OR = 4.308, CI = [3.190–5.819] and *p* < 0.001). Female smokers showed significant high-risk association with the GG genotype and the *G. allele* ((OR = 101 GG, 167.42 G); (CI = [5.905–1717.54, *p* = 0.001); (CI = [10.137–276509], *p* = 0.001), respectively). The SNPs showed a high-risk association with age for both those under and those above 28 years, as shown in Table 3D. The GG variant (homozygous) in smokers under 28 years showed a high-risk relationship (OR = 2.316; CI = [1.507–3.229]; *p* < 0.001). Furthermore, a high-risk association was observed in the *G. allele* (OR = 2.038; CI = [1.544–2.691]; *p* < 0.001). Furthermore, the GG and G variants in smokers above 28 years showed significant association. The GG homozygous variant genotype has a high-risk relationship with smokers in subjects older than 28 years (OR = 3.0434; CI = [1.8937–4.8912]; *p* < 0.001). Additionally, the *G. allele* showed a significant high-risk association with smokers older than 28 years (OR = 2.582; CI = [1.891–3.527]; *p* < 0.001).

SNP rs3176752 showed no significant association with clinical parameters, including CS duration, daily CS average, gender, and younger smokers, as shown in Table 3A–D.

#### 3.3.2. Estimation for SNPs XPC (rs1870134, rs2228000, rs2228001, and rs2607775)

The genotyping and allele distributions of SNPs *XPA* rs1870134 and rs2228000 were estimated in order to investigate the link between clinical parameters and the risk of smoking-induced diseases. These SNPs showed no significant association with all clinical parameters. The comparison of alleles and genotyping frequencies between subjects with the four clinical characteristics did not show any correlation because the *p* value is not statistically significant.

The analysis result in SNP rs2228001 does not show a significant association with the risk of smoking causing disease, considering genotyping and allele frequencies and statistical values, except for the female gender. The CC genotype and the C allele have a high-risk association with smoking-induced diseases ((OR = 18.765 CC, 2.810 C); CI = [1.012–347.841], [1.246–6.377], *p* = 0.044, 0.013, respectively).

SNP 2607775CC presented a high-risk association with all clinical parameters. Based on the C allele and the CC genotype frequencies comparison, the SNP between subjects with short-term smokers and long-term smokers compared to non-smokers showed significant high-risk association with smoking duration (Table 4A). Similarly, the CC and the *C. allele* genotype displayed a significant high-risk relationship with moderate smokers when compared to non-smokers (see Table 4B). Additionally, the SNP showed significant high-risk association with both male and female smokers (Table 4C). Lastly, there was a significant high-risk association for age above 28 between smokers and non-smokers (Table 4D).

### 3.4. Observed and Expected Counts

The null hypothesis is that there is no difference in the genotypes or alleles and the results in the equilibrium. So, if the *p* value is <0.05, we will reject the null hypothesis, and the genotype/alleles will be in disequilibrium (Table 5).

## 4. Discussion

CS is a major cause of various serious diseases, and several studies have shown that the cause of these diseases is due to tobacco chemicals [27]. For decades, scientific studies have not considered the damaging effects of CS on the human body, such as the lungs, respiratory system, and oral cavity. Nevertheless, there is evidence that CS promotes inflammation in the oral cavity and assists the development of gingival and periodontal disease by stimulating the secretion of inflammatory cytokines [28]. These findings illustrate the possibility of a powerful and effective approach to learning more about the effects of CS on oral mucosae. Previous studies showed that there are different types of cancers, and diseases are related to CS. Indeed, CS could affect many cellular pathways, such as KRAS and P53, causing certain defects, and it may increase the risk of cancer and disease [29]. For instance, it has been found that CS induces DNA damage and has an impact on the genes controlling the cell cycle and DNA replication and repair. Mutations or polymorphisms in these genes were found to be linked with smoking and various type of cancer [4,7,30]. Furthermore, it was reported that HIF-1α expression in human NSCLC cells was induced by exposure to Nicotine in s CS [31]. HIF was associated with GLUT-1 in hypoxia conditions. Elevated GLUT1 expression has been associated with many cancers, including hepatic, pancreatic, renal, breast, ovarian, brain, esophageal, lung, cutaneous, endometrial, colorectal, and cervical [32].

The risks of CS were underestimated, since those risks were assumed to be well linked, with machine-determined CS yields. However, the changes in people’s smoking behaviors were not accounted for in machine-determined smoke yields [33]. Studies estimating exposure biomarkers would have presented more accurate assessments of risk, as these biomarkers will be a result of smoking behavior, as well the characteristics of the cigarette itself [34]. To better understand the association between CS and relative diseases, it is, therefore, critical to evaluate how polymorphisms function in related genes. For example, *XPA* and *XPC* genes may contribute to the disease because of CS. Due to the serious health condition caused by smoking in the Saudi population; this study could indicate the early effects of the smoking-induced disease because of genetic variations to several affected genes following the CS exposure. This shows the importance of evaluating the effects of CS on causing disease or cancer by looking into associations with a genetic variation on the number of genes in different pathways.

The present study aimed to investigate the *XPA* and *XPC* gene polymorphisms’ variations in the NER pathway, an important part of the DNA repair system damage caused by CS in smokers versus non-smokers of the Saudi population, to detect a genetic marker that could help predict disease, thus reducing the risks caused by CS among healthy individuals. In this work, four SNPs were selected (rs10817938, rs1800975, rs3176751, and rs3176752) and distributed in different regions of *XPA* gene, and four SNPs (rs1870134, rs2228000, rs2228001, and rs2607775) were distributed in different regions of the *XPC* gene. There have been many studies suggesting that *XPA* and *XPC* polymorphisms had a significant effect on the risk of cancer and disease, and they could be a biomarker [35,36].

Furthermore, the associations were validated between *XPA* and *XPC* SNPs and clinical characteristics, including CS duration, daily CS average, gender, and age. The *XPA* and *XPC* SNPs appeared to be significantly affected by CS, resulting in genetic changes in the DNA repair system gene. Given that, the *XPA* polymorphisms are related to the risk of many types of cancers [21,37]. In this study, SNP rs10817938 results showed a significant low-risk association only with the duration of smoking. SNPs rs1800975 and rs3176751 results showed significant low-risk and high-risk associations, respectively, with regard to all clinical parameters. However, rs3176752 showed no significant association in all parameters. The results of *XPA* rs3176751 polymorphisms might increase the influence of these clinical parameters regarding disease caused by CS. It is considered that CS contains chemical carcinogens that are known to produce genetic mutations that may not be repaired by the NER pathway because this mutation may not be recognized by *XPA* with the rs3176751 mutant gene type. The study results of SNP 10817938 for the Saudi population are inconsistent with the study that confirmed the association of *XPA* polymorphisms with oral squamous cell carcinoma (OSCC) risk in the Han Chinese population. The results demonstrated a significant high-risk association between CS and CC homozygous genotype in rs10817938 of OSCC, *p* < 0.01, OR = 3.60 [38]. For rs1800975, the results of this study are similar to a prior study, demonstrating that this variant was associated with a significantly reduced risk of lung cancer [39]. Although the percentage of female smokers used in this study was very low (13.1%) compared to the male smokers (59.9%), it was intriguing to find a significant high-risk difference in the genotypic and the allelic distribution of *XPA* rs3176751 in female smokers, suggesting a possible interference of CS in disease development among women, as reported previously for innate immune genes in acute respiratory distress syndrome [40] and human papillomavirus [41]. The results of the study indicate that gender may have a significant role in the association between the *XPA* rs3176751 polymorphism and the cancer risk or other diseases.

A comparison between smokers with clinical characteristics to controls revealed that there were no associations observed between SNPs rs1870134 and rs2228000 in the *XPC* gene and smokers. These results do not match the previous study, which showed that *XPC* rs1870134 was verified to be correlated with a decreased risk of hepatocellular and prostate cancers [42,43]. Additionally, SNP rs2228000 CT/TT genotype revealed a protective effect of gastric cancer only significant among subjects older than 58 years in a Southern Chinese population [44]. However, there were significant high-risk associations between the rs2228001 polymorphism and female smokers. SNP rs2607775 showed significant high-risk association with all clinical parameters.

Finally, this work offers various strengths and benefits. One of its strengths is that this study determined polymorphisms in the *XPA* and *XPC* genes in three categories of the SNP site, including 3′ UTR, 5′ UTR, and exon variants. Second, all samples were obtained from the same region of Riyadh and not from different regions of Saudi Arabia, and they were carefully monitored and stored according to protocol. However, due to the social traditions of our community, this study was limited with regards to the adequacy of samples from female smokers.

## 5. Conclusions

Overall, the present study results demonstrated possible significant associations between CS and SNPs polymorphisms in DNA repair genes, such as *XPA* and *XPC*, and these effects of polymorphism can be a key factor in the development of CS-induced disease. The exact mechanism of how smoking influences genetic changes that cause cancer or disease remains unclear. Therefore, future studies are required to investigate the expressions of *XPA* and *XPC* gene and the link between polymorphisms and the rate of CS. Additionally, we suggest examining the oxidatively generated guanine lesion 8-oxoguanin to evaluate the oxidative stress DNA damage that occurs by CS. A further investigation comparing our results with other previously studied populations involving different ethnicities and CS habits may help define the effects of CS on different genes involved in DNA repair systems. The finding of identified SNPs polymorphisms associated with the CS induced disease could be used as biomarkers.

## Figures and Tables

**Table 1 genes-14-01349-t001:** Demographic characteristics of 421 smokers and non-smokers in the study.

Characteristics	All *n* = 421	Smokers *n* = 201	Non-Smokers *n* = 220	*p* Values
Gender, % Male Female	314 (74.6%) 107 (25.4%)	187 (59.9%) 14 (13.1%)	127 (40.1%) 93 (86.9%)	<0.001
Age, mean (±SD)	27.8 (7.3)	28.1 (6.1)	27.6 (8.3)	0.413
Smoking frequency per day, % ≤10 per day >10 per day Not specified	77 (18.3%) 97 (23.0%) 27 (6.4%)	77 (38.3%) 97 (48.3%) 27 (13.4%)		
Duration of smoking in years, % ≤5 years >5 years Not specified	65 (15.4%) 112 (26.6%) 24 (5.7%)	65 (32.3%) 112 (55.7%) 24 (11.9%)		

*n:* Number, SD: Standard deviation. *p*-Value in bold represents significant result.

**Table 2 genes-14-01349-t002:** General genotype frequencies of four *XPA* and four *XPC* gene polymorphisms in smoker cases and controls.

SNP	Variant	Smoker	Non-Smoker	Chi-Square *p* Values	OR (CI)
*XPA* gene					
rs10817938	TT TC CC	7 (3.7%) 69 (36.7%) 112 (59.6%)	12 (6.0%) 47 (23.6%) 140 (70.4%)	0.071 0.521	Ref. 2.517 (0.923–6.863) 1.371 (0.523–3.599)
	TC+CC T C	181 (96.3%) 83 (22.1%) 293 (77.9%)	187 (94.0%) 71 (18.0%) 327 (82.0%)	0.868 Ref. 0.141	1.025 (0.770–1.363) Ref. 0.767 (0.538–1.092)
rs1800975	TT TC CC	186 (96.4%) 6 (3.1%) 1 (0.5%)	119 (55.1%) 14 (6.5%) 83 (38.4%)	**0.009 *** **<0.001 ***	Ref. 0.274 (0.103–0.733) 0.008 (0.001–0.056)
	TC+CC T C	7 (3.6%) 378 (97.9%) 8 (2.1%)	97 (45.97%) 252 (58.3%) 180 (41.7%)	**<0.001 *** Ref. **<0.001 ***	0.081 (0.037–0.178) Ref. 0.029 (0.014–0.061)
rs3176751	CC CG GG	65 (33.3%) 17 (8.7%) 113 (58.0%)	151 (69.0%) 25 (11.4%) 43 (19.6%)	0.188 **<0.001 ***	Ref. 1.579 (0.799–3.122) 6.105 (3.869–9.631)
	CG+GG G C	130 (66.7%) 147 (37.7%) 243 (62.3%)	68 (31.1%) 327 (74.7%) 111 (25.3%)	**<0.001 *** Ref. **<0.001 ***	2.147 (1.511–3.050) Ref. 4.869 (3.618–6.555)
rs3176752	GG GT TT	195 (97.0%) 6 (3.0%) 0	208 (96.0%) 8 (4.0%) 0	0.0685	Ref. 0.800 (0.273–2.347)
	GT+TT G T	6 (2.99%) 396 (98.5%) 6 (1.5%)	8 (3.7%) 424 (98.1%) 8 (1.9%)	0.094 Ref. 0.687	0.806 (0.275–2.363) Ref. 0.803 (0.276–2.335)
*XPC* gene					
rs1870134	GG GC CC	1 (0.5%) 7 (3.7%) 184 (95.8%)	0 13 (6.0%) 205 (94.0%)	0.319 0.461	Ref. 0.185 (0.007–5.137) 0.299 (0.012–7.392)
	GC+CC G C	191 (99.5%) 9 (2.3%) 375 (97.7%)	218 (100%) 13 (3.0%) 423 (97.0%)	0.970 Ref. 0.574	0.995 (0.756–1.309) Ref. 1.281 (0.541–3.029)
Rs2228000	CC CT TT	3 (1.5%) 45 (23.2%) 146 (75.3%)	2 (0.9%) 49 (22.9%) 163 (76.2%)	0.600 0.575	Ref. 0.612 (0.098–3.834) 0.597 (0.098–3.624)
	CT+TT C T	191 (98.0%) 51 (13.1%) 337 (86.9%)	212 (99.0%) 53 (12.4%) 375 (87.6%)	0.965 Ref. 0.745	0.994 (0.754–1.309) Ref. 0.934 (0.619–1.409)
rs2228001	AA AC CC	64 (34.4%) 82 (44.1%) 40 (21.5%)	72 (34.5%) 95 (45.4%) 42 (20.1%)	0.898 0.805	Ref. 0.971 (0.620–1.519) 1.071 (0.619–1.854)
	AA+CC A C	122 (65.6%) 210 (56.5%) 162 (43.5%)	137 (65.6%) 239 (57.2%) 179 (42.8%)	0.932 Ref. 0.837	0.987 (0.726–1.341) Ref. 1.030 (0.777–1.366)
rs2607775	GG GC CC	79 (41.8%) 72 (38.1%) 38 (20.1%)	111 (51.4%) 86 (39.8%) 19 (8.8%)	0.455 0.001	Ref. 1.176 (0.769–1.801) 2.810 (1.509–5.233)
	GC+CC G C	110 (58.2%) 230 (60.8%) 148 (39.2%)	105 (48.6%) 308 (71.3%) 124 (28.7%)	0.287 Ref. **0.002 ***	1.197 (0.859–1.667) Ref. 1.598 (1.192–2.143)

Note: * number of valid cases specified in each of the SNPs, missing data were excluded from the analysis. SNP: Single nucleotide polymorphism, Ref = Reference allele, OR: Odd ratio, CI = Confidence interval. *p*-Values in bold represent significant results.

**Table 3 genes-14-01349-t003:** Comparison of *XPA* genotypes and allele distributions according to the duration of smoking, frequency of smoking, and gender.

*A. Based on the duration of smoking*								
SNP	Variant	Control	Duration of smoking		Chi-square *p* values		OR (CI)	
			≤5 years	>5 years	Control vs. <5 years	Control vs. >5 years	Control vs. <5 years	Control vs. >5 years
rs10817938	TT TC CC	12 (6.0%) 47 (23.6%) 140 (70.4%)	3 (3.7%) 28 (34.1%) 51 (62.2%)	1 (1.2%) 29 (34.5%) 54 (64.3%)	Ref. 0.207 0.572	Ref. 0.061 0.146	Ref. 2.383 (0.618–9.182) 1.457 (0.395–5.374)	Ref. 7.404 (0.914–59.975) 4.629 (0.587–36.463)
	TC+CC T C	187 (94.0%) 71 (18.0%) 327 (82.0%)	31 (37.8%) 37 (22.2%) 130 (77.8%)	30 (35.7%) 31 (18.5%) 137 (91.5%)	**<0.001 *** Ref. 0.235	**<0.001 *** Ref. 0.862	0.402 (0.254–0.637) Ref. 0.763 (0.488–1.192)	0.380 (0.239–0.603) Ref. 0.960 (0.602–1.530)
rs1800975	TT TC CC	119 (55.1%) 14 (6.5%) 83 (38.4%)	84 (96.6%) 2 (2.3%) 1 (1.1%)	80 (95.2%) 4 (4.8%) 0	Ref. **0.038 *** **0.049 ***	Ref. 0.144 **0.009 ***	Ref. 0.202 (0.045–0.914) 0.084 (0.072–0.994)	Ref. 0.425 (0.135–1.338) 0.019 (0.001–0.145)
	TC+CC T C	97 (45.97%) 252 (58.3%) 180 (41.7%)	3 (3.4%) 170 (97.7%) 4 (2.3%)	4 (4.8%) 164 (97.6%) 4 (2.4%)	**<0.001 *** Ref. **<0.001 ***	**<0.001 *** Ref. **<0.001 ***	0.077 (0.024–0.249) Ref. 0.033 (0.012–0.090)	0.106 (0.037–0.297) Ref. 0.034 (0.012–0.093)
rs3176751	CC CG GG	151 (69.0%) 25 (11.4%) 43 (19.6%)	25 (28.4%) 7 (8.0%) 56 (63.6%)	29 (34.1%) 9 (10.6%) 47 (55.3%)	Ref. 0.273 **<0.001 ***	Ref. 0.152 **<0.001 ***	Ref. 1.691 (0.661–4.325) 7.866 (4.402–14.057)	Ref. 1.874 (0.794–4.427) 5.691 (3.207–10.099)
	CG+GG C G	68 (31.1%) 327 (74.7%) 111 (25.3%)	63 (71.6%) 57 (32.4%) 119 (67.6%)	56 (65.9%) 67 (39.4%) 103 (60.6%)	**<0.001 *** Ref. **<0.001 ***	**0.001 *** Ref. **<0.001 ***	2.305 (1.510–3.518) Ref. 6.150 (4.197–9.013)	2.121 (1.376–3.273) Ref. 4.529 (3.111–6.593)
rs3176752	GG GT TT	208 (96.0%) 8 (4.0%) 0	88 (100%) 0 0	85 (94.4%) 5 (5.6%) 0	Ref. 0.161 0.694	Ref. 0.544 0.684	Ref. 0.129 (0.007–2.264) 2.197 (0.043–111.66)	Ref. 1.426 (0.453–4.487) 2.275 (0.044–115.60)
	GT+TT G T	8 (3.7%) 424 (98.1%) 8 (1.9%)	0 176 (100%) 0	5 (5.6%) 175 (97.2%) 5 (2.8%)	0.169 Ref. 0.180	0.562 Ref. 0.472	0.134 (0.007–2.357) Ref. 0.142 (0.008–2.465)	1.402 (0.446–4.406) Ref. 1.514 (0.489–4.693)
*B. Based on frequency of smoking*								
SNP	Variant	Control	Frequency of smoking per day		Chi-square *p* values		OR for cases (CI)	
			≤10	>10	Control vs. ≤10	Control vs. >10	Control vs. ≤10	Control vs. >10
rs10817938	TT TC CC	12 (6.0%) 47 (23.6%) 140 (70.4%)	1 (1.5%) 24 (35.3%) 43 (63.2%)	3 (3.2%) 32 (34.0%) 59 (62.8%)	Ref. 0.090 0.216	Ref. 0.144 0.432	Ref. 6.128 (0.752–49.963) 3.686 (0.466–29.164)	Ref. 2.723 (0.711–10.427) 1.686 (0.459–6.193)
	TC+CC T C	187 (94.0%) 71 (18.0%) 327 (82.0%)	67 (98.5%) 26 (19.1%) 110 (80.9%)	91 (96.8%) 41 (21.5%) 150 (78.5%)	0.813 Ref. 0.739	0.868 Ref. 0.294	1.048 (0.708–1.552) Ref. 0.919 (0.558–1.512)	1.030 (0.725–1.463) Ref. 0.794 (0.517–1.222)
rs1800975	TT TC CC	119 (55.1%) 14 (6.5%) 83 (38.4%)	68 (93.2%) 4 (5.5%) 1 (1.4%)	92 (97.9%) 2 (2.1%) 0	Ref. 0.238 **0.001 ***	Ref. **0.028** **<0.001 ***	Ref. 0.500 (0.158–1.579) 0.016 (0.002–0.114)	Ref. 0.185 (0.041–0.834) 0.008 (0.001–0.126)
	TC+CC T C	97 (45.97%) 252 (58.3%) 180 (41.7%)	5 (6.8%) 140 (96.6%) 5 (3.4%)	2 (2.1%) 186 (98.9%) 2 (1.1%)	**0.001 *** Ref. **<0.001 ***	**<0.001 *** Ref. 0.738	0.152 (0.059–0.389) Ref. 0.050 (0.020–0.124)	0.047 (0.011–0.196) Ref. 1.400 (0.195–10.032)
rs3176751	CC CG GG	151 (69.0%) 25 (11.4%) 43 (19.6%)	21 (28.8% 6 (8.2%) 46 (63.0%)	31 (32.3%) 10 (10.4%) 55 (57.3%)	Ref. 0.285 **<0.001 ***	Ref. 0.115 **<0.001 ***	Ref. 1.725 (0.634–4.696) 7.692 (4.148–14.262)	Ref. 1.948 (0.850–4.463) 6.230 (3.575–10.858)
	CG+GG C G	68 (31.1%) 327 (74.7%) 111 (25.3%)	52 (71.2%) 48 (32.9%) 98 (67.1%)	65 (97.7%) 72 (37.5%) 120 (62.5%)	**0.001 *** Ref. **<0.001 ***	**0.001 *** Ref. **<0.001 ***	2.294 (1.466–3.590) Ref. 6.015 (4.004–9.035)	2.180 (1.438–3.305) Ref. 4.909 (3.415–7.058)
rs3176752	GG GT TT	208 (96.0%) 8 (4.0%) 0	73 (94.8%) 4 (5.2%) 0	96 (99.0%) 1 (1.0%) 0	Ref. 0.573 nan	Ref. 0.221 nan	Ref. 1.425 (0.417–4.872) nan	Ref. 0.271 (0.033–2.196) nan
	GT+TT G T	8 (3.7%) 424 (98.1%) 8 (1.9%)	4 (5.2%) 146 (97.3%) 4 (2.7%)	1 (1.0%) 193 (99.5%) 1 (0.5%)	0.589 Ref. 0.547	0.231 Ref. 0.225	1.403 (0.411–4.789) Ref. 1.452 (0.431–4.893)	0.278 (0.034–2.256) Ref. 0.275 (0.034–2.211)
*C. Based on gender*								
SNP	Variant	Control	Gender		Chi-square *p* values		OR (CI)	
			Male	Female	Control vs. Male	Control vs. Female	Control vs. Male	Control vs. Female
rs10817938	TT TC CC	12 (6.0%) 47 (23.6%) 140 (70.4%)	7 (4.0%) 67 (38.5%) 100 (57.5%)	0 2 (14.3%) 12 (85.7%)	Ref. 0.081 0.681	Ref. 0.862 0.587	Ref. 2.444 (0.895–6.669) 1.225 (0.466–3.220)	Ref. 1.316 (0.059–29.198) 2.224 (0.124–39.835)
	TC+CC T C	187 (94.0%) 71 (18.0%) 327 (82.0%)	167 (96.0%) 111 (29.4%) 267 (70.6%)	14 (100%) 2 (7.1%) 26 (92.9%)	0.887 Ref. 0.158	0.874 Ref. 0.164	1.021 (0.763–1.367) Ref. 0.522 (0.372–0.733)	1.064 (0.491–2.292) Ref. 2.823 (0.655–12.166)
rs1800975	TT TC CC	119 (55.1%) 14 (6.5%) 83 (38.4%)	175 (97.8%) 3 (1.7%) 1 (0.6%)	11 (78.6%) 3 (21.4%) 0	Ref. **0.003 *** **<0.001 ***	Ref. 0.236 0.056	Ref. 0.146 (0.041–0.518) 0.008 (0.001–0.059)	Ref. 2.318 (0.577–9.322) 0.006 (0.004–1.071)
	TC+CC T C	97 (45.97%) 252 (58.3%) 180 (41.7%)	4 (2.2%) 353 (98.6%) 5 (1.4%)	3 (21.4%) 25 (89.3%) 3 (10.7%)	**<0.001 *** Ref. **<0.001 ***	0.253 Ref. **0.004 ***	0.049 (0.018–0.137) Ref. 0.019 (0.008–0.018)	0.477 (0.134–1.698) Ref. 0.168 (0.050–0.565)
rs3176751	CC CG GG	151 (69.0%) 25 (11.4%) 43 (19.6%)	65 (35.9%) 17 (9.4%) 99 (54.7%)	0 0 14 (100%)	Ref. 0.188 **<0.001 ***	Ref. **0.018 *** **0.001 ***	Ref. 1.580 (0.799–3.122) 5.349 (3.372–8.482)	Ref. 5.941 (0.115–306.217) 101.00 (5.905–1717.54)
	CG+GG C G	68 (31.1%) 327 (74.7%) 111 (25.3%)	116 (64.1%) 147 (40.6%) 215 (59.4%)	14 (100%) 0 28 (100%)	**0.001 *** Ref. **<0.001 ***	**0.004 *** Ref. **0.001 ***	2.064 (1.442–2.953) Ref. 4.308 (3.190–5.819)	3.221 (1.463–7.091) Ref. 167.42 (10.137–2765-09)
rs3176752	GG GT TT	208 (96.0%) 8 (4.0%) 0	181 (96.8%) 6 (3.2%) 0	14 (100%) 0 0	Ref. 0.787 nan	Ref. 0.910 nan	Ref. 0.862 (0.294–2.531) nan	Ref. 0.846 (0.047–15.395) nan
	GT+TT G T	8 (3.7%) 424 (98.1%) 8 (1.9%)	6 (3.2%) 368 (98.4%) 6 (1.6%)	0 28 (100%) 0	0.794 Ref. 0.789	0.930 Ref. 0.928	0.866 (0.295–2.542) Ref. 0.864 (0.297–2.514)	0.878 (0.048–15.984) Ref. 0.876 (0.049–15.567)
*D. Based on age*								
SNP	Variant	Control	age		Chi-square		OR (CI)	
			≤ 28 years	>28 years	≤28 years vs. control	≤28 years vs. control	≤28 years	>28 years vs. control
rs10817938	TT TC CC	12 (6.0%) 47 (23.6%) 140 (70.4%)	12 (5.0%) 62 (26.1%) 164 (68.9%)	7 (4.8%) 52 (35.9%) 86 (59.3%)	Ref. 0.540 0.709	Ref. 0.215 0.917	Ref. 1.319 (0.544–3.198) 1.171 (0.510–2.690)	Ref. 1.897 (0.689–5.219) 1.053 (0.399–2.778)
	TC+CC T C	187 (94.0%) 71 (18.0%) 327 (82.0%)	226 (95.0%) 86 (18.1%) 390 (81.9%)	138 (95.2%) 66 (22.8%) 224 (77.2%)	0.940 Ref. 0.930	0.935 Ref. 0.111	1.011 (0.771–1.324) Ref. 0.985 (0.696–1.393)	1.013 (0.745–1.377) Ref. 0.737 (0.506–1.073)
rs1800975	TT TC CC	119 (55.1%) 14 (6.5%) 83 (38.4%)	175 (68.6%) 15 (5.9%) 65 (25.5%)	126 (84.0%) 5 (3.3%) 19 (12.7%)	Ref. 0.417 **0.002 ***	Ref. **0.043 *** **<0.001 ***	Ref. 0.729 (0.339–1.565) 0.533 (0.357–0.794)	Ref. 0.337 (0.118–0.965) 0.216 (0.124–0.378)
	TC+CC T C	97 (45.97%) 252 (58.3%) 180 (41.7%)	80 (31.4%) 365 (71.6%) 145 (28.4%)	24 (16.0%) 257 (85.7) 43 (14.3%)	**0.043 *** Ref. <**0.001 ***	**<0.001 *** Ref. <**0.001 ***	0.699 (0.493–0.989) Ref. 0.556 (0.424–0.729)	0.356 (0.218–0.583) Ref. 0.234 (0.161–0.341)
rs3176751	CC CG GG	151 (69.0%) 25 (11.4%) 43 (19.6%)	138 (53.5%) 29 (11.2%) 91 (35.3%)	75 (49.3%) 12 (7.9%) 65 (42.8%)	Ref. 0.422 **<0.001 ***	Ref. 0.928 **<0.001 ***	Ref. 1.269 (0.709–2.273) 2.316 (1.507–3.229)	Ref. 0.966 (0.460–2.029) 2.316 (1.507–3.559)
	CG+GG C G	68 (31.1%) 327 (74.7%) 111 (25.3%)	120 (46.5%) 305 (59.1%) 211 (40.9%)	77 (50.7%) 162 (53.3%) 142 (46.7%)	**0.023 *** Ref. **<0.001 ***	**0.013 *** Ref. **<0.001 ***	1.498 (1.058–2.121) Ref. 2.038 (1.544–2.691)	1.632 (1.109–2.401) Ref. 2.582 (1.891–3.527)
rs3176752	GG GT TT	208 (96.0%) 8 (4.0%) 0	250 (96.9%) 8 (3.1%) 0	149 (96.1%) 6 (3.9%) 0	Ref. 0.718 nan	Ref. 0.936 nan	Ref. 0.832 (0.307–2.255) nan	Ref. 1.047 (0.356–3.081) nan
	GT+TT G T	8 (3.7%) 424 (98.1%) 8 (1.9%)	8 (3.1%) 508 (98.4%) 8 (1.6%)	6 (3.9%) 304 (98.1%) 6 (1.9%)	0.727 Ref. 0.720	0.936 Ref. 0.934	0.837 (0.309–2.268) Ref. 0.835 (0.311–2.243)	1.045 (0.355–3.073) Ref. 1.046 (0.359–3.046)

* = *p* < 0.05, SNP: Single nucleotide polymorphism, Ref = Reference allele, OR: Odd ratio, CI = Confidence interval. *p*-Values in bold represent significant results.

**Table 4 genes-14-01349-t004:** Comparison of *XPC* genotypes and alleles distribution according to duration of smoking, frequency of smoking, and gender.

*A. Based on duration of smoking*								
SNP	Variant	Control	Duration of smoking		Chi-square *p* values		OR (CI)	
			≤5 years	>5 years	Control vs. <5 years	Control vs. >5 years	Control vs. <5 years	Control vs. >5 years
rs1870134	GG GC CC	0 13 (6.0%) 205 (94.0%)	1 (1.1%) 4 (4.5%) 83 (94.3%)	0 1 (1.2%) 82 (98.8%)	Ref. 0.202 0.222	Ref. 0.313 0.649	Ref. 0.111 (0.003–3.243) 0.135 (0.006–3.359)	Ref. 0.111 (0.002–7.927) 0.402 (0.008–20.403)
	GC+CC G C	218 (100%) 13 (3.0%) 423 (97.0%)	87 (98.8%) 6 (3.4%) 170 (96.6%)	83 (100%) 1 (0.6%) 165 (99.4%)	0.949 Ref. 0.783	1.000 Ref. 0.119	0.989 (0.696–1.404) Ref. 0.871 (0.326–2.329)	1.000 (0.699–1.430) Ref. 5.071 (0.658–39.084)
Rs2228000	CC CT TT	2 (0.9%) 49 (22.9%) 163 (76.2%)	0 18 (20.7%) 69 (79.3%)	3 (3.5%) 24 (28.2%) 58 (68.2%)	Ref. 0.691 0.628	Ref. 0.237 0.120	Ref. 1.868 (0.085–40.783) 2.125 (0.100–44.849)	Ref. 0.326 (0.051–2.086) 0.237 (0.038–1.456)
	CT+TT C T	212 (99.0%) 53 (12.4%) 375 (87.6%)	87 (100%) 18 (10.3%) 156 (89.7%)	82 (96.5%) 30 (17.6%) 140 (82.4%)	0.831 Ref. 0.482	0.787 Ref. 0.095	0.962 (0.675–1.371) Ref. 1.225 (0.695–2.158)	1.052 (0.728–1.519) Ref. 0.660 (0.405–1.075)
rs2228001	AA AC CC	72 (34.5%) 95 (45.4%) 42 (20.1%)	25 (29.4%) 43 (50.6%) 17 (20.0%)	31 (37.8%) 31 (37.8%) 20 (24.4%)	Ref. 0.371 0.678	Ref. 0.353 0.771	Ref. 1.304 (0.729–2.329) 1.166 (0.565–2.405)	Ref. 0.758 (0.422–1.360) 1.106 (0.561–2.181)
	AA+CC A C	137 (65.6%) 239 (57.2%) 179 (42.8%)	42 (49.4%) 93 (52.5%) 77 (47.5%)	51 (62.2%) 93 (56.7%) 71 (43.3%)	0.195 Ref. 0.584	0.802 Ref. 0.918	0.754 (0.491–1.156) Ref. 1.106 (0.772–1.583)	0.949 (0.629–1.431) Ref. 1.019 (0.708–1.468)
rs2607775	GG GC CC	111 (51.4%) 86 (39.8%) 19 (8.8%)	32 (37.6%) 32 (37.6%) 21 (24.7%)	36 (43.9%) 29 (35.4%) 17 (20.7%)	Ref. 0.376 **0.001 ***	Ref. 0.892 **0.008 ***	Ref. 1.291 (0.734–2.271) 3.834 (1.839–7.993)	Ref. 1.040 (0.591–1.828) 2.759 (1.297–5.868)
	GC+CC G C	105 (48.6%) 308 (71.3%) 124 (28.7%)	53 (62.4%) 96 (56.5%) 74 (43.5%)	46 (56.1%) 101 (61.6%) 63 (38.4%)	0.855 Ref. **0.001 ***	0.514 Ref. **0.023 ***	1.038 (0.693–1.556) Ref. 1.915 (1.325–2.766)	1.154 (0.751–1.774) Ref. 1.549 (1.062–2.260)
*B. Based on frequency of smoking*								
SNP	Variant	Control	Frequency of smoking per day		Chi-square *p* values		OR (CI)	
			≤10	>10	Control vs. ≤10	Control vs. >10	Control vs. ≤10	Control vs. >10
rs1870134	GG GC CC	0 13 (6.0%) 205 (94.0%)	0 6 (8.5%) 65 (91.5%)	1 (1.0%) 1 (1.0%) 94 (97.9%)	Ref. 0.722 0.569	Ref. 0.074 0.252	Ref. 0.482 (0.009–27.092) 0.319 (0.006–16.223)	Ref. 0.037 (0.001–1.380) 0.153 (0.006–3.798)
	GC+CC G C	218 (100%) 13 (3.0%) 423 (97.0%)	71 (100%) 6 (4.2%) 136 (95.8%)	95 (99.0%) 3 (1.6%) 189 (98.4%)	1.00 Ref. 0.473	0.952 Ref. 0.307	1.00 (0.685–1.461) Ref. 0.697 (0.260–1.868)	0.989 (0.704–1.390) Ref. 1.936 (0.545–6.874)
Rs2228000	CC CT TT	2 (0.9%) 49 (22.9%) 163 (76.2%)	1 (1.4%) 22 (30.1%) 50 (68.5%)	2 (2.1%) 19 (20.0%) 74 (77.9%)	Ref. 0.932 0.693	Ref. 0.360 0.434	Ref. 0.898 (0.077–10.433) 0.614 (0.055–6.908)	Ref. 0.388 (0.051–2.953) 0.454 (0.063–3.285)
	CT+TT C T	212 (99.0%) 53 (12.4%) 375 (87.6%)	72 (98.6%) 24 (16.4%) 122 (83.6%)	93 (97.9%) 23 (12.4%) 163 (87.6%)	0.982 Ref. 0.216	0.946 Ref. 0.995	0.996 (0.683–1.451) Ref. 0.718 (0.426–1.213)	0.988 (0.701–1.393) Ref. 1.002 (0.594–1.689)
rs2228001	AA AC CC	72 (34.5%) 95 (45.4%) 42 (20.1%)	20 (27.8%) 35 (48.6%) 17 (23.6%)	33 (36.3%) 39 (42.9%) 19 (20.9%)	Ref. 0.379 0.325	Ref. 0.698 0.970	Ref. 1.326 (0.707–2.488) 1.457 (0.688–3.086)	Ref. 0.896 (0.514–1.561) 0.987 (0.499–1.949)
	AA+CC A C	137 (65.6%) 239 (57.2%) 179 (42.8%)	37 (51.4%) 75 (52.1%) 69 (47.9%)	52 (57.1%) 105 (57.7%) 77 (52.3%)	0.290 Ref. **<0.001 ***	0.505 Ref. 0.954	0.784 (0.499–1.231) Ref. 2.569 (1.643–4.019)	0.872 (0.583–1.305) Ref. 1.011 (0.707–1.443)
rs2607775	GG GC CC	111 (51.4%) 86 (39.8%) 19 (8.8%)	23 (31.9%) 27 (37.5%) 22 (30.6%)	43 (47.3%) 33 (36.3%) 15 (16.5%)	Ref. 0.191 **<0.001 ***	Ref. 0.972 0.067	Ref. 0.515 (0.812–2.826) 5.588 (2.612–11.956)	Ref. 0.991 (0.581–1.689) 2.038 (0.950–4.371)
	GC+CC G C	105 (48.6%) 308 (71.3%) 124 (28.7%)	50 (69.4%) 73 (50.7%) 71 (49.3%)	76 (83.5%) 119 (65.4%) 63 (34.6%)	0.104 Ref. **<0.001 ***	**0.006 *** Ref. 0.147	1.429 (0.929–2.195) Ref. 2.416 (1.639–3.559)	1.718 (1.171–2.521) Ref. 1.315 (0.909–1.903)
*C. Based on gender*								
SNP	Variant	Control	Gender		Chi-square *p* values		OR (CI)	
			Male	Female	Control vs. Male	Control vs. Female	Control vs. Male	Control vs. Female
rs1870134	GG GC CC	0 13 (6.0%) 205 (94.0%)	1 (0.6%) 6 (3.4%) 171 (96.1%)	0 1 (7.1%) 13 (92.9%)	Ref. 0.282 0.434	Ref. 0.313 0.178	Ref. 0.161 (0.006–4.505) 0.278 (0.012–6.873)	Ref. 0.111 (0.002–7.927) 0.066 (0.001–3.441)
	GC+CC G C	218 (100%) 13 (3.0%) 423 (97.0%)	177 (99.4%) 8 (2.2%) 348 (97.8%)	14 (100%) 1 (3.4%) 27 (96.6%)	0.969 Ref. 0.524	1.00 Ref. 0.859	0.994 (0.751–1.316) Ref. 1.337 (0.548–3.262)	1.00 (0.466–2.147) Ref. 0.829 (0.105–6.582)
Rs2228000	CC CT TT	2 (0.9%) 49 (22.9%) 163 (76.2%)	3 (1.7%) 40 (22.2%) 137 (76.1%)	0 5 (35.7%) 9 (64.3%)	Ref. 0.516 0.529	Ref. 0.716 0.435	Ref. 0.544 (0.087–3.412) 0.560 (0.092–3.402)	Ref. 0.556 (0.024–13.116) 0.291 (0.013–6.488)
	CT+TT C T	212 (99.0%) 53 (12.4%) 375 (87.6%)	177 (98.3%) 46 (12.8%) 314 (87.2%)	14 (100%) 5 (17.9%) 23 (82.1%)	0.959 Ref. 0.868	0.981 Ref. 0.403	0.993 (0.749–1.315) Ref. 0.965 (0.632–1.472)	1.009 (0.469–2.168) Ref. 0.650 (0.237–1.783)
rs2228001	AA AC CC	72 (34.5%) 95 (45.4%) 42 (20.1%)	64 (37.2%) 73 (42.4%) 35 (20.3%)	0 9 (64.3%) 5 (35.7%)	Ref. 0.527 0.986	Ref. 0.068 **0.044 ***	Ref. 0.862 (0.544–1.366) 1.005 (0.566–1.758)	Ref. 14.381 (0.823–251.404) 20.093 (1.081–373-437)
	AA+CC A C	137 (65.6%) 239 (57.2%) 179 (42.8%)	99 (57.6%) 163 (53.3%) 143 (46.7%)	5 (35.7%) 9 (32.1%) 19 (67.9%)	0.437 Ref. 0.296	0.254 Ref. **0.013 ***	0.878 (0.633–1.219) Ref. 1.171 (0.871–1.579)	0.545 (0.192–1.547) Ref. 2.819 (1.246–6.377)
rs2607775	GG GC CC	111 (51.4%) 86 (39.8%) 19 (8.8%)	78 (44.6%) 69 (39.4%) 28 (16.0%)	1 (7.1%) 3 (21.4%) 10 (71.4%)	Ref. 0.545 **0.026 ***	Ref. 0.245 **0.001 ***	Ref. 1.142 (0.743–1.754) 2.097 (1.094–4.019)	Ref. 3.872 (0.396–37.881) 58.421 (7.065–483.07)
	GC+CC G C	105 (48.6%) 308 (71.3%) 124 (28.7%)	106 (60.6%) 225 (64.3%) 125 (35.7%)	13 (92.8%) 5 (17.9%) 23 (82.1%)	0.198 Ref. **0.037 ***	0.108 Ref. **<0.001 ***	1.246 (0.891–1.743) Ref. 1.379 (1.020–1.867)	1.910 (0.867–4.209) Ref. 11.426 (4.284–30.729)
*D. Based on age*								
SNP	Variant	Control	age		Chi-square		OR (CI)	
			≤ 28 years	>28 years	≤ 28 years vs. control	>28 years vs. control	≤ 28 years vs. control	>28 years vs. control
rs1870134	GG GC CC	0 13 (6.0%) 205 (94.0%)	0 12 (4.7%) 243 (95.3%)	1 (0.7%) 8 (5.3%) 142 (94.0%)	Ref. 0.969 0.933	Ref. 0.356 0.371	Ref. 0.926 (0.017–50.291) 1.185 (0.023–59.983)	Ref. 0.209 (0.008–5.769) 0.231 (0.009–5.715)
	GC+CC G C	218 (100%) 13 (3.0%) 423 (97.0%)	255 (100%) 12 (2.4%) 498 (97.6%)	150 (99.4%) 10 (3.3%) 292 (96.7%)	1.00 Ref. 0.549	0.965 Ref. 0.800	1.0 (0.774–1.291) Ref. 1.275 (0.576–2.825)	0.993 (0.741–1.333) Ref. 0.897 (0.388–2.074)
Rs2228000	CC CT TT	2 (0.9%) 49 (22.9%) 163 (76.2%)	5 (2.0%) 56 (22.0%) 193 (76.0%)	0 37 (24.7%) 113 (75.3%)	Ref. 0.362 0.376	Ref. 0.395 0.423	Ref. 0.457 (0.085–2.463) 0.474 (0.091–2.474)	Ref. 3.788 (0.177–81.266) 3.471 (0.165–72.987)
	CT+TT C T	212 (99.0%) 53 (12.4%) 375 (87.6%)	249 (98.0%) 66 (13.0%) 442 (87.0%)	150 (100%) 37 (12.3%) 263 (87.7%)	0.937 Ref. 0.781	0.950 Ref. 0.984	0.989 (0.764–1.281) Ref. 0947 (0.643–1.394)	1.009 (0.751–1.356) Ref. 1.005 (0.645–1.573)
rs2228001	AA AC CC	72 (34.5%) 95 (45.4%) 42 (20.1%)	91 (37.0%) 107 (43.5%) 48 (19.5%)	42 (29.0%) 69 (47.6%) 34 (23.4%)	Ref. 0.586 0.703	Ref. 0.381 0.277	Ref. 0.891 (0.589–1.349) 0.904 (0.539–1.516)	Ref. 1.245 (0.762–2.034) 1.388 (0.769–2.506)
	AA+CC A C	137 (65.6%) 239 (57.2%) 179 (42.8%)	139 (56.5%) 289 (58.7%) 203 (41.3%)	76 (52.4%) 153 (52.8%) 137 (47.2%)	0.331 Ref. 0.634	0.212 Ref. 0.245	0.862 (0.639–1.163) Ref. 0.938 (0.720–1.221)	0.799 (0.5562–1.136) Ref. 1.196 (0.885–1.616)
rs2607775	GG GC CC	111 (51.4%) 86 (39.8%) 19 (8.8%)	130 (51.4%) 87 (34.4%) 36 (14.2%)	57 (38.5%) 70 (47.3%) 21 (14.2%)	Ref. 0.463 0.123	Ref. **0.044 *** **0.031 ***	Ref. 0.864 (0.584–1.277) 1.618 (0.878–2.979)	Ref. 1.585 (1.012–2.486) 2.152 (1.071–4.325)
	GC+CC G C	105 (48.6%) 308 (71.3%) 124 (28.7%)	123 (48.6%) 347 (68.6%) 159 (31.4%)	91 (62.3%) 184 (62.2%) 112 (37.8%)	0.999 Ref. 0.366	0.188 Ref. **0.010 ***	1.001 (0.728–1.374) Ref. 1.138 (0.859–1.507)	1.265 (0.891–1.795) Ref. 1.512 (1.104–2.069)

* = *p* < 0.05, SNP: Single nucleotide polymorphism, Ref = Reference allele, OR: Odd ratio, CI = Confidence interval. *p*-Values in bold represent significant results.

**Table 5 genes-14-01349-t005:** Hardy-Weinberg Equilibrium Test and for all genotypes and alleles in XPA and XPC genes.

		Nonsmokers		Hardy-Weinberg Chi-Square Equilibrium Test	Smokers		Hardy-Weinberg Chi-Square Equilibrium Test	Interpretation
		Observed Expected	Expected		Observed	Expected		
rs10817938	TT TC CC	12 47 140	9 59.6 129.6	5.959 *p* = 0.051	7 69 112	9.2 56.4 122.4	5.435 *p* = 0.057	Equilibrium
rs1800975	TT TC CC	119 14 83	161.1 10.6 44.4	5.865 *p* = 0.053	186 6 1	143.9 9.4 39.6	5.911 *p* = 0.052	Equilibrium
rs3176751	CC CG GG	151 25 43	114.3 22.2 82.5	1.812 *p* = 0.404	65 17 113	101.7 19.8 73.5	1.806 *p* = 0.405	Equilibrium
rs3176752	GG GT	208 8	208.7 7.3	13.312 *p* < 0.001 *	195 6	194.3 6.7	13.517 *p* < 0.001 *	disEquilibrium
rs1870134	GG GC CC	0 13 205	5 10.6 205.8	19.081 *p* < 0.001 *	1 7 184	5 9.4 182.2	17.438 *p* < 0.001 *	disEquilibrium
Rs2228000	CC CT TT	2 49 163	2.6 49.3 162.1	26.323 *p* < 0.001 *	3 45 146	2.4 44.7 146.9	25.831 *p* < 0.001 *	disEquilibrium
rs2228001	AA AC CC	72 95 42	72 93.7 43.4	0.318 *p* = 0.853	64 82 40	64 83.3 38.6	0.317 *p* = 0.853	Equilibrium
rs2607775	GG GC CC	111 86 19	101.3 84.3 30.4	0.933 *p* = 0.627	79 72 38	88.7 73.7 26.6	0.933 *p* = 0.627	Equilibrium

* = *p* < 0.05. *p*-Values in bold represent significant results

## Data Availability

All data generated or analyzed during this study are included in this published article.

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
