# Peer review of "A Significant Increasing Risk Association between Cigarette Smoking and XPA and XPC Genes Polymorphisms"

_genes, 2023, doi:10.3390/genes14071349_

Round 1

Reviewer 1 Report

The manuscript describes the frequencies of 08 polymorphisms on XPA and XPC genes among smokers and healthy controls. Evaluated SNPs are located in genetic regions on genes that could produce alterations in the DNA repair system. According to the Authors, these variants may be associated with the nucleotide excision repair pathways caused by cigarette smoking in Saudi population. However, the results obtained could be improved with some methodological and analytical features that must be addressed:

1.            The Authors did not calculate sample size by their SNPs’ minor allele frequency and expected OR to be addressed.

2.            Genetic polymorphisms may vary in their frequency among different populations. The Authors did not describe if their cases/controls were from a specific geographical location, race, or ethnic group that could explain the difference in frequency.

3.            The Smokers group in Table 1 did not match to the n = 200 with Gender (188 + 13 = 201), Period of Smoking (86 + 91 = 177), and Average of CS/ Day (74 + 99 = 173).

4.            The Authors showed 11 tables for the same subjects with different variable categorizations. Th

5.            Hardy-Weinberg equilibrium was only described by the Authors in the Statistical analysis section, but it was not reported/discussed in the Results.

6.            There are differences between groups in Gender and could be on Age, although there were not calculated in the study.

7.            The Authors did not analyze linkage disequilibrium on their SNPs. This could highlight their results.

8.            The Authors used the terms Moderate and Heavy smokers. However, they did not ever categorize or defined them.

9.            The Authors used a dominant genetic model without an a priori hypothesis. Usually, authors had a prior scope of which genetic model will be useful according to the literature background.

I suggest that the manuscript could be reviewed by a native speaker in order to correct some redaction, wording issues [e.g. “As decscribed by our previousely” (page 03, line 122), “gende, and ages” (page 07, line 203); “This SNPs” (page 11, line 275); “Firs,”(page 24, line 450). In addition, in the discussion section, there were no strengths or limitations addressed 

Reviewer 2 Report

Many genes could be altered and mutated by smoking other than KRAS and p53, such as GLUT-1, HIF-1, p16 . They should be mentioned in the introduction. There is a relationship between  ERCC-1 and sensitivity to cisplatin therapy, please state this.

Please include in the methods additional information about the study population and inclusion/exclusion criteria. In particular, data regarding pack-years , second-hand smoke exposure, the possible use of e-cigarette, comorbidities, staging should be entered. Were former smokers in the control group? Was logistic regression analysis included? please explain

What smoke related diseases risks were considered?

Please include a legend in table 2 and 3

The interference of smoking with EGFR-TKI and chemotherapy should be included in the discussion. Please insert a paragraph about the perspectives of these findings.

Please include the following references useful for discussion:

-A review. Thorac Cancer. 2020 Nov;11(11):3060-3070. 

- Cancer. 2017 Aug 23;8(14):2846-2853.

-Future Sci OA. 2019 May 3;5(5):FSO394. 
